# The Influence of Community Feeders and Commercial Food Outlets on the Spatial Distribution of Free-Roaming Dogs—A Photographic Capture and Recapture Study

**DOI:** 10.3390/ani13050824

**Published:** 2023-02-24

**Authors:** Saulo Nascimento de Melo, Eduardo Sérgio da Silva, Renata Aparecida Nascimento Ribeiro, Paulo Henrique Araújo Soares, Anna Karolyna Rodrigues Cunha, Cláudia Maria de Souza Gonçalves, Flávia Daniela Santos Melo, Marco Aurélio Pereira Horta, Rafael Gonçalves Teixeira-Neto, Vinícius Silva Belo

**Affiliations:** 1Campus Centro-Oeste Dona Lindu, Universidade Federal de São João del-Rei, Divinópolis 35501-296, MG, Brazil; 2Campus Divinópolis, Universidade do Estado de Minas Gerais, Divinópolis 35501-170, MG, Brazil; 3Fundação Oswaldo Cruz, Instituto Oswaldo Cruz, Rio de Janeiro 21040-900, RJ, Brazil

**Keywords:** photographic captures, stray dogs, ecology and behavior, animal welfare

## Abstract

**Simple Summary:**

Interactions between free-roaming dogs and humans influence the quality of life and behavior of both species. Understanding the spatial distribution of free-roaming dogs is essential in designing policies to control zoonoses and improve canine well-being. In the present study, by means of photographic captures and recaptures and geospatial position recordings of 554 dogs, we demonstrated that the location of the animals in an urban environment was influenced by the direct supply of food offered by the human population. Free-roaming dogs stayed closer to community feeders than to commercial food outlets. We know that community feeders are essential to improve the quality of life of free-roaming dogs. However, they should be in areas with reduced movement of people/vehicles. Our results may be representative of different areas of Brazil and of other parts of the world. They expand the understanding of canine ecology and behavior in the urban environment and highlight the importance of human contributions to the maintenance and distribution of free-roaming dogs.

**Abstract:**

Understanding the distribution of dogs in the environment is relevant for establishing human and animal health actions. In the present study, we analyzed the influence of community feeders and commercial food outlets on the spatial distribution of free-roaming dogs in an urban area of a municipality in Southeast Brazil. The dogs were identified via photographic capture and recapture performed over five sampling efforts. The spatial densities of dogs were determined using the Kernel method. Spatial correlations between the distribution of free-roaming dogs and the locations of community feeders and commercial food outlets were analyzed using the K function. During the study, 1207 captures/recaptures were performed encompassing 554 dogs, the majority (62.6%) of which were males. Agglomerations of male and female dogs were observed in the areas where food was present. Positive spatial autocorrelations were detected between the distribution of dogs and food sources. The median distances between dogs and community feeders or commercial food outlets were 1.2 and 1.4 km, respectively, and the difference between these two was statistically significant. The presence of community feeders and food outlets demonstrates the influence of human activity, on the spatial distribution of free-roaming dogs. These results will be useful for developing strategies aimed at the improvement of animal welfare and the prevention of zoonoses.

## 1. Introduction

In Brazil, dogs are part of the urban ecosystem in both small country towns and large city conurbations [1]. The canine population within the country is estimated to number around 54.2 million animals [2], with a large proportion being free-roaming dogs that roam the streets freely without the direct supervision of humans [3,4].

Although dogs provide numerous benefits to humans, such as companionship, stress reduction, physical activity, improved mental health, assistance for people with disabilities, protection, and safety, free-roaming animals cause accidents and spread infectious diseases [5,6,7]. Moreover, such free-roaming dogs may have a shorter lifespan in comparison with domiciled animals [8,9], considering that the former receive minimal or no veterinary care [10] and have limited access to adequate nutrition. For these reasons, the management of free-roaming dog populations is gaining increased research attention and is considered a relevant issue from the viewpoint of public health and animal welfare [11].

While interactions between restricted dogs and humans exhibit many different forms [3], the density of the free-roaming canine population tends to increase concurrently with that of the human population. The maintenance of these animals in urban environments depends on direct sources of food and support (community feeders), and on indirect sources such as commercial food outlets (stores and restaurants) and garbage collection points [12,13]. The availability of food influences the spatial distribution of dogs, while the search for food motivates them to travel longer distances within their environs [14,15,16]. The population dynamics of free-roaming dogs are influenced substantially by the supply of food by householders since a small number of community feeders can support a large canine population [10,13,17].

Despite its likely relevance to the ecology and spatial distribution of free-roaming dogs within the environment, the direct supply of food by the human population has been poorly analyzed in the literature. A better understanding of this issue may be pertinent for population control, improvement of animal welfare and the prevention of zoonoses [18,19]. In a study carried out previously in the same area as the present work, agglomerations of unrestricted dogs were identified in places close to food stores [19]. This study extends the investigation of the influence of the human community on the ecology of the canine population. In the present investigation, we set out to analyze the influence of community feeders and commercial food outlets on the ecology and spatial distribution of free-roaming canine populations in an urban setting by carrying out photographic captures and recaptures.

## 2. Materials and Methods

### 2.1. Site of Study

The investigation was performed in Divinópolis, the largest municipality in the mid-west of the state of Minas Gerais, Brazil, with a population of approximately 242,505 inhabitants [20]. The specific study site encompassed eight neighborhoods in the municipality comprising some 7600 residents [9,21].

### 2.2. Data Collection

Photographic captures and recaptures of free-roaming dogs (defined as animals roaming the streets not accompanied by an owner) were performed over five sampling efforts (campaigns) conducted between September 2018 and September 2019 inclusive. Samplings were carried out in the mornings of three consecutive days during the first two weeks of the month and were repeated every three months. The vehicle used during each sampling always followed the same route and covered all of the streets in the designated areas at a speed of 20 km h^−1^. The team comprised the driver together with two researchers who were responsible for photographing and recording the general characteristics of the photographed animals, including size, color, natural marks and sex, all of which were useful for later identification. A Canon PowerShot SX60HS camera was used to photograph the dogs at a maximum distance of 10 m in order to obtain good-quality pictures while taking care not to frighten the animals away. At least three photographs of each dog were taken from different positions. The geographical coordinates of each capture/recapture point (the place where the dog was sighted) were recorded together with those of community feeders (food such as dry dog food and food scraps provided by households either in containers or directly on the ground) and commercial food outlets (fast-foods, grocery stores, bars, restaurants, bakeries, butchers and food markets).

Data concerning dogs captured during the three days of sampling were aggregated in order to assemble the database from the five campaigns. The aggregation of data in campaigns was defined to increase the robustness and power of comparisons and, as well as to compare our results with that of past literature.

Dogs that had been photographed in one campaign but captured again in a new campaign were considered recaptured. Each animal received an identification code and its details (pictures, description and position) were recorded in a spreadsheet. Data were analyzed and classified independently by two researchers. Using the Kappa index, a high (0.86) and significant (*p* < 0.01) agreement was recorded between them in identifying the 1207 photocaptures. Any discrepancies were resolved by consensus. Finally, a database was created containing information about all of the dogs captured and/or recaptured during the five campaigns.

### 2.3. Statistical Analysis

The nearest-neighbor distance approach was used to identify the existence of regions with agglomerations of dogs in the study area [22]. Kernel maps were prepared for the entire study period, and separately for each capture/recapture event, and stratified by sex of the dogs. The search radius was set at 100 m and the quantile normalization method was adopted as the most appropriate statistical tool for visualization of the different concentrations of dogs in the geographic space [23]. Possible spatial correlations between the distribution of free-roaming dogs and the locations of community feeders and commercial food outlets were investigated using Ripley’s bivariate K function [19,24]. The linear distance matrix function was used to estimate the distances of the meeting points of the dogs in relation to the community feeders and commercial food outlets, with the values expressed as medians and interquartile ranges (IQR). Median distances were compared using the Mann–Whitney test with the significance level set at 5%. Spatial analyses were performed using QGIS software version 3.16.16 while statistical analyses were carried out with the aid of R software version 4.2.0.

## 3. Results

A total of 1207 photographic captures/recaptures involving 554 different free-roaming dogs were accomplished during the study period (September 2018 and September 2019 inclusive). The proportion of captured free-roaming male dogs was typically twice that of females, with a similar profile observed for recaptured dogs. (Table 1). Most dogs in all campaigns were small, as compared to medium and large animals. In addition, black dogs were predominant (Table 2). The number of dogs captured for the first time decreased as capture efforts progressed. The number of recaptured individuals tended to increase over the study period, although there was a decrease in the last effort. (Figure 1).

The numbers of community feeders and commercial food outlets increased slightly from the first to the third sampling effort but decreased thereafter. There were more community feeders than commercial food outlets (Table 3).

According to nearest neighbor distance analysis, the animals were distributed in a clustered manner, that is, they were not randomly distributed in geographic space. The distribution patterns of agglomeration were statistically significant (*p* < 0.01). As shown by the Kernel map (Figure 2), animal clusters were located in the vicinity of community feeders and food outlets. Although the sites of food sources varied over the study period, canine agglomerations always remained close to these locations (Appendix A) regardless of the sex of the animals (Appendix A).

The K function revealed the existence of positive spatial autocorrelations up to 500 m between the distribution of free-roaming dogs and food sources (Figure 3). The median distance between dogs and community feeders was 1.2 km (IQR = 0.5–2.6 km), whereas the median distance between dogs and commercial food outlets was significantly higher at 1.4 km (IQR = 0.5–2.7 km; *p* < 0.01).

## 4. Discussion

The results demonstrate that the interaction between humans and dogs plays a relevant role in the distribution of the animals in the study area with the occurrence of clusters being positively correlated with the presence of food sources, particularly community feeders.

The greater number of male dogs and the agglomeration of males and females in the vicinity of food outlets in the study area has been established previously by Melo et al. (2020) [19] with distance values similar to those reported herein. The predominance of males occurs due to behavioral factors of the dogs and cultural aspects related to the care of animals in the region [19]. The consistency of the results related to the agglomeration may be explained by the absence of interventions aimed at the dispersion of the animals and the lack of control of the free-roaming canine population [25]. In addition, the number of new free-roaming dogs captured decreased as the number of dogs recaptured increased. The existence of new free-roaming dogs (captured) in the region after a year of continuous captures, may demonstrate the lack of responsible custody by owners (canine abandonment or escape) and, to a lesser extent, the existence of canine reproduction [9].

The novel finding of dog clusters in areas close to community feeders reinforces the significance of human-dog interactions in the maintenance of free-roaming dogs in urban settings. It is of note that during the study, a small number of dog clusters were observed in locations that had no feeders or food outlets. This finding can be explained by the behavior of some guardians who allow free access of their domiciled dogs to the streets and even feed them outside their properties [26]. Such results show that in the absence of interventions that consider the responsible ownership of animals and the home ranges of dogs, there will be no reduction in the stray dog population.

Dogs, unlike wolves from which they evolved, depend primarily on humans for food [27,28]. Thus, direct support by community feeders through the provision of both food and care for the free-roaming animals favors the continuation and augmentation of the free-roaming canine population in the urban environment [9]. Thus, human support is essential for understanding the ecology of unrestrained dogs in urban environments. Families who regularly feed free-roaming dogs are responsible for sustaining these animals, so only a small fraction of households can accommodate large, tolerant, unrestrained dogs [10,13].

In the present study, free-roaming dogs gathered together closer to community feeders than to food outlets. Most of the time there were no humans near the feeders, a situation that contrasts with that established for food outlets. Kittisiam et al. (2021) analyzed the contact network of free-roaming dogs in a university campus in Thailand and demonstrated that the average number of contacts for the weekend network was significantly higher than that for the weekday network, indicating that dogs tended to cluster more intensely in the absence of humans.

Our study showed that community feeders might be more advantageous to the animals than food outlets because they afford a continuous and possibly more stable nutritional source. As part of an educational project (‘Projeto AlimentaCão’) aimed at alerting schoolchildren to the problem of abandonment of domestic animals, [29] and installed feeders at points on the campus of the Universidade Tecnolólogica Federal do Paraná that were some distance away from the university restaurant. These researchers reported that the number of free-roaming dogs that congregated in the surroundings of the restaurant diminished whilst the population using the feeder points stabilized and became healthier and more docile. An alternative approach of restricting the amount of food would also be effective in controlling a population of free-roaming dogs at a particular location [30]. However, such a measure would be ethically questionable and would simply lead to the dispersion of animals as they move to new areas in search of food [9,25]. The more advantageous solution would be to relocate community feeders to defined areas with less movement of people and vehicles. Although the control of dog habitats and their ranges may not result in a decrease in the canine population, it may be effective in reducing accidents, the transmission of diseases, bites and pollution in busier locations.

The availability of food resources was constant throughout the period of the present study, a situation that may be representative of different areas of Brazil and of other parts of the world. In India, for example, 37% of individuals reportedly feed free-roaming dogs [17]. Feeding free-roaming dogs can create and strengthen the bonds between humans and dogs [31], and allow the animals to move, socialize and express their natural behavior [32]. On the other hand, the support offered by community feeders can also be considered a public health problem because it sustains canine populations that act as reservoirs of zoonotic diseases [33,34], cause accidents and pollute the environment. Moreover, from the viewpoint of animal welfare, the quality of life and the longevity of free-roaming dogs tend to be inferior to those of domiciled animals [35,36].

While various studies support the thesis that the search for food is a key factor in determining canine mobility and agglomeration [19,37,38], it should be noted that other factors influence the distribution of free-roaming dogs in urban areas including the growth of cities, climate, reproductive status and the search for partners [39,40,41]. Therefore, controlling the habitat and movement of free-roaming dogs is a challenging task that must be adopted together with other strategies. Sterilization can contribute to the reduction in birth rates, but in the region of the present study, the abandonment of dogs is the main reason for the increase in the unrestricted population [9].

In order to reduce or eliminate the population of free-roaming dogs, it will be necessary to adopt broad and effective measures of responsible animal guardianship, which may be difficult to implement in the short term depending on the socioeconomic and cultural contexts [26,42]. In this sense, the provision of food and the management of appropriate care may be relevant actions. Installing feeders in favorable environments can reduce the risk of accidents and bites, minimize the risk of disease transmission and improve animal welfare [43]. In addition, if combined with other actions such as vaccination, disease prevention and installation of shelters, the provision of feeders would certainly improve the quality of life of the animals [44,45]. However, it is important to emphasize that such measures are palliative and that the problems associated with free-roaming dogs will only be solved through the awareness and accountability of guardians concerning the social and economic consequences caused by the abandonment of their animal companions [46], along with the implementation of responsible adoption policies [26,42].

Strategies to estimate animal populations share as basic principles, uncertainties regarding the detection of all individuals that pertain to the target population in a given area and heterogeneities in the individual encounter probabilities [47]. The validity of the methods based on counting or the capture and recapture of dogs is only achieved when the animals are correctly identified in all capture activities [48]. Obtaining accurate information about free-roaming dogs can be a challenging process, and inadequate methods can result in biased information [26,47]. Among the various methods for estimating population, photographic capture and recapture is advantageous because it is a safe, fast, and cost-effective option. Animals do not need to be physically captured, reducing their exposure to risks and adverse effects. Moreover, the photographic capture technique enables the individual identification of animals based on natural markings, and analysis of the previous capture history can determine whether an animal has been recaptured [38]. Combining this method with the Geographical Coordinate System (GPS) to record the location of free-roaming dogs allows for expanded analysis and a better understanding of the ecology of unrestricted dogs in urban areas [21]. On the other hand, the low quality and lack of detail in the photograph can make the process of identifying the dog more difficult. However, in the present study, the comparison of photos by independent researchers, combined with quality training, can optimize the identification process.

The present study was subject to some limitations that are intrinsic to the photographic capture method. The majority of free-roaming dogs move around, which makes it difficult to photograph them, and this may give rise to the loss of some information. However, during the search process, all streets in the study area were covered and this minimized sample deficiencies because when a dog could not be found in one street, it was almost certain that it could be found in another. An additional limitation was that the sampling efforts took place only in the morning and not throughout the day. However, dogs are more active in the morning and tend to rest as the temperature rises [49]; therefore, sampling efforts were focused in the period of highest activity.

## 5. Conclusions

Agglomerations of free-roaming dogs were positively associated with the locations of commercial food outlets and, more especially, with those of community feeders. The provision of food by community feeders was essential in maintaining the concentration of dogs in the vicinity of food sources. The results obtained not only add to our understanding of the ecology and behavior of free-roaming dogs in urban areas but also highlight the need for policies to reduce the risks associated with the presence of free-roaming animals. Moreover, appropriate management of food and water resources should be implemented in order to improve the quality of life of these animals such as the allocation of specific and supervised points far from busy roads and residential/commercial areas. We hope that the outlined recommendations can help improve future interventions by providing more appropriate strategies. Additionally, we suggest that the spatial distribution of free-roaming dogs be more widely studied in the literature, especially through intervention studies, assessments in urban and rural areas, and evaluations of residents’ perceptions of supporting these animals.

## Figures and Tables

**Figure 1 animals-13-00824-f001:**
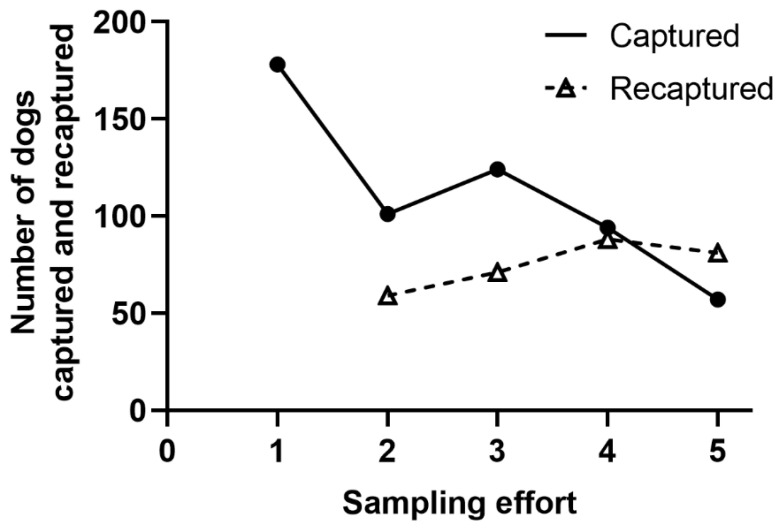
Temporal evolution in the number of free-roaming dogs captured/recaptured during the five sampling efforts carried out in an urban area located in Divinópolis, Minas Gerais, Brazil.

**Figure 2 animals-13-00824-f002:**
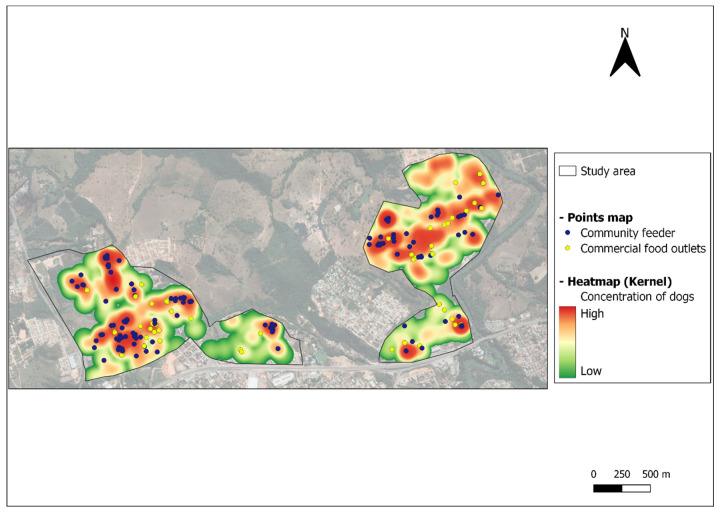
Clusters of free-roaming dogs and sites of community feeders and commercial food outlets in an urban area located in Divinópolis, Minas Gerais, Brazil.

**Figure 3 animals-13-00824-f003:**
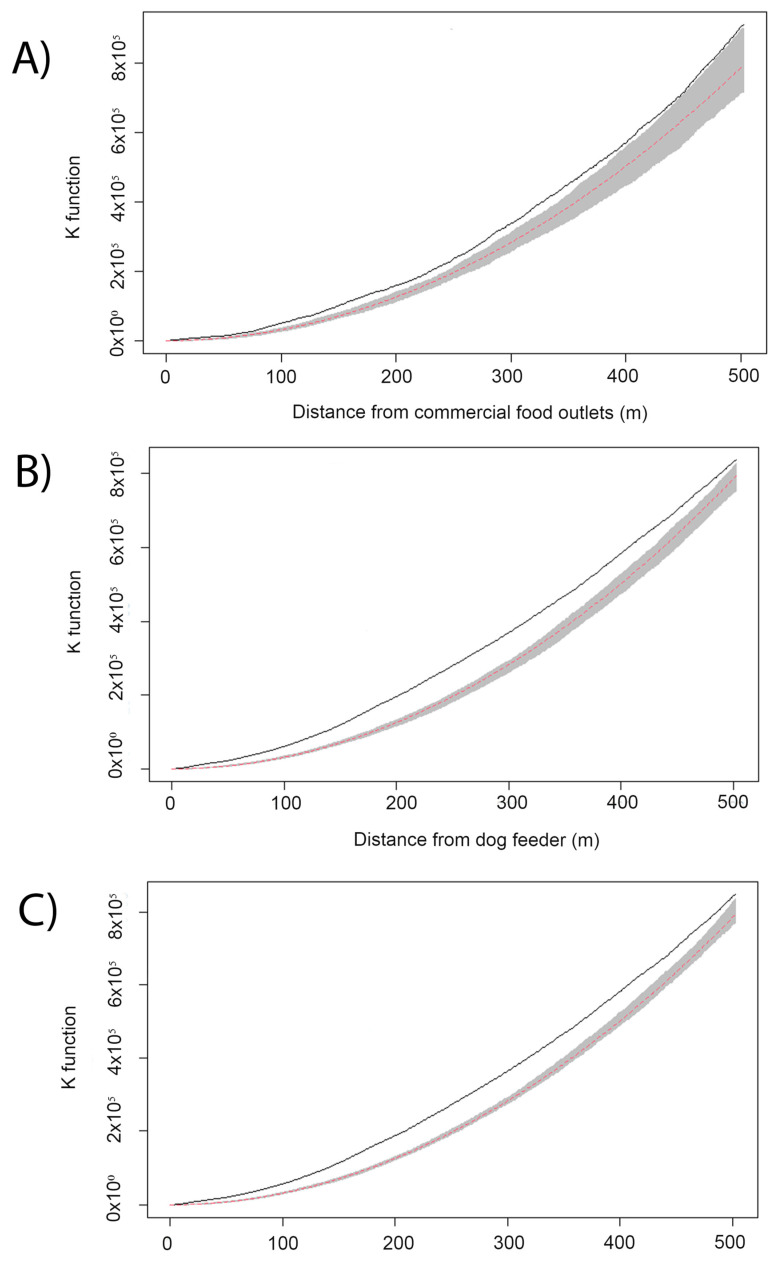
The positive spatial autocorrelations between the distribution of free-roaming dogs and (**A**) commercial food outlets, (**B**) community feeders and (**C**) community feeders + commercial food outlets, showing the theoretical Poisson K-function (dashed red line), the envelope of confidence (gray band) and the observed K function (solid black line).

**Table 1 animals-13-00824-t001:** Number of free-roaming dogs captured/recaptured during the five sampling efforts carried out in an urban area located in Divinópolis, Minas Gerais, Brazil.

Animals	Sampling Efforts					
	1st	2nd	3rd	4th	5th	Total
**Captured dogs [*n* (%)]**						
Males	111 (62.3)	71 (70.3)	77 (62.0)	51 (54.3)	37 (65.0)	347 (62.6)
Females	63 (35.4)	27 (26.7)	38 (30.7)	40 (42.5)	18 (31.5)	186 (33.6)
Undetermined	4 (2.3)	3 (3.0)	9 (7.3)	3 (3.2)	2 (3.5)	21 (3.8)
Total	178	101	124	94	57	554
**Recaptured dogs [*n* (%)]**						
Males	-	41 (69.5)	50 (70.4)	55 (67.9)	55 (67.9)	279 (64.7)
Females	-	18 (30.5)	20 (28.2)	32 (36.4)	25 (30.9)	142 (33.0)
Undetermined	-	0 (0)	1 (1.4)	2 (2.2)	1 (1.2)	9 (2.2)
Total	-	59	71	88	81	431

**Table 2 animals-13-00824-t002:** Characteristics of free-roaming dogs captured during the five sampling efforts carried out in an urban area located in Divinópolis, Minas Gerais, Brazil.

Sampling Efforts						
Captured Dogs [*n* (%)]	1st	2nd	3rd	4th	5th	**Total**
**Size**						
Small	98 (54.7)	55 (54.5)	70 (56.5)	66 (70.2)	31 (54.4)	320 (57.7)
Medium	41 (23.0)	25 (24.8)	26 (21.0)	13 (13.8)	16 (28.1)	121 (21.8)
Big	39 (22.3)	21 (20.7)	28 (22.5)	15 (16.0)	10 (17.5)	113 (20.5)
**Total**	178	101	124	94	57	554
**Color**						
Yellow	25 (14.0)	10 (9.9)	6 (4.8)	3 (3.2)	4 (7.0)	48 (8.6)
White	35 (19.6)	18 (17.8)	26 (21.0)	17 (18.1)	11 (19.3)	107 (19.3)
Gray	2 (1.1)	0 (0.0)	5 (4.0)	3 (3.2)	1 (1.8)	11 (2.0)
Brown	39 (21.9)	22 (21.8)	40 (32.3)	33 (35.1)	22 (38.6)	156 (28.1)
Black	78 (43.8)	51 (50.5)	47 (37.9)	38 (40.4)	19 (33.3)	233 (42.0)
Total	178	101	124	94	57	554

**Table 3 animals-13-00824-t003:** Number of food sources identified during the five sampling efforts carried out in an urban area located in Divinópolis, Minas Gerais, Brazil.

Sampling Efforts	Community Feeders	Commercial Food Outlets	Total
1st	25	17	42
2nd	30	31	61
3rd	36	32	68
4th	27	27	54
5th	33	17	50
Total	151	124	275

## Data Availability

The data presented in this study are available in the Appendix A of this study.

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
