# Peer review of "The Influence of Community Feeders and Commercial Food Outlets on the Spatial Distribution of Free-Roaming Dogs—A Photographic Capture and Recapture Study"

_animals, 2023, doi:10.3390/ani13050824_

Round 1

Reviewer 1 Report

This is clear written paper describing interactions between free-roaming dogs and humans with respect the influence of community feeders and commercial food outlets on the ecology and spatial distribution of free-roaming canine population. The study was comprehensively designed and clearly explained. Free-roaming dogs stayed closer to community feeders than to commercial food outlets.

The results contributed to understanding of canine ecology and behavior in the  urban environment and highlight the importance of the human contribution in the maintenance and  distribution of free-roaming dogs.

Author Response

 We would like to thank the reviewer for her/his kind comments.

Reviewer 2 Report

Congratulations on the job!

The only suggestion I make is to include a legend in FIGURE 3. Understanding is possible without the legend, but it would help a lot in understanding the graphs.

Author Response

- Thanks for the comment. Figure 3 has a caption in lines 163-168.

“Figure 3. The positive spatial autocorrelations between the distribution of free-roaming dogs and (A) commercial food outlets, (B) community feeders and (C) community feeders + commercial food outlets, showing the theoretical Poisson K-function (dashed red line), the envelope of confidence (gray band) and the observed K function (solid black line)”.

Reviewer 3 Report

This study starts with the issue of free-roaming dogs and the food resources that affects their distribution.  The study aims to collect information on rural dogs' demography for future provisions against dog-borne disease transmission. In the results, the distribution of dogs has been revealed however the discussion was a bit far from my expectations. 

I prefer to ask you to expand the discussion for your results which leads to controlling the dog's habitat and their ranges. Furthermore, the ideal or target number of dogs should be described together with the local government's policy if applicable.

Author Response

We would like to thank the reviewer for these important suggestions. Based on them, we have expanded the discussion about the control of the dogs’ habitat and their ranges (see below). On the other hand, we prefer not to define the number of dogs that would be acceptable. There is no specific legislation for this in Brazil and this issue may vary depending on the sociocultural context of the localities.

Lines 190-192: “Such results show that in the absence of interventions that consider the responsible ownership of animals and the home ranges of dogs, there will be no reduction in the stray dog population”.

Lines 217-219: “Such results show that in the absence of interventions that consider the responsible ownership of animals and the home ranges of dogs, there will be no reduction in the stray dog population.”

Lines 233-237: “Therefore, control the habitat and movement of free-roaming dogs is a challenging task that must be adopted together with other strategies. Sterilization can contribute to the reduction of birth rates, but in the region of the present study, the abandonment of dogs is the main reason for the increase in the unrestricted population [9].”

Reviewer 4 Report

 The main question was  where do stray dogs congregate. The topic is not very original, but extends it to another place. The contribution of community feeders to urban dog ecology is interesting. The conclusions are consistent with the evidence. The references are extensive. Dogs recaptured only once or twice 554 dogs and 1207 photos. Describe  food given by community (meat ? dry dog food?) and type of restaurant  (fast food vs fine dining) of the 26 or so food outlets. Any explanation of sex difference? Any explanation for precipitous fall in number of dogs. from 200 to 50 

Because you have recorded the information, a table of size hair length and coat color  should be included

What is the difference between a campaign and a day? There should have been 18 days of data

Table 

Round 2

Reviewer 3 Report

The author responded to my comment appropriately.

I only concern about the arrangement of tables in the manuscripts.

Author Response

Thanks for the comment.
The arrangement of the tables has been checked and is in line with Animals standards.